# Nurse–Patient Communication and Relationship When Wearing Personal Protective Equipment: Nurses’ Experience in a COVID-19 Ward

**DOI:** 10.3390/healthcare11131960

**Published:** 2023-07-07

**Authors:** Raffaella Gualandi, Dhurata Ivziku, Rosario Caruso, Chiara Di Giacinto, Marzia Lommi, Daniela Tartaglini, Anna De Benedictis

**Affiliations:** 1Department of Health Professions, Fondazione Policlinico Universitario Campus Bio-Medico, 00128 Rome, Italy; r.gualandi@policlinicocampus.it (R.G.); chiaradigiacinto1998@gmail.com (C.D.G.); d.tartaglini@policlinicocampus.it (D.T.); 2Clinical Research Service, IRCCS Policlinico San Donato, San Donato Milanese, 20097 Milano, Italy; rosario.caruso@unimi.it; 3Department of Biomedical Sciences for Health, University of Milan, 20133 Milano, Italy; 4Department of Biomedicine and Prevention, University Tor Vergata, 00133 Rome, Italy; marzia.lommi@gmail.com; 5Clinical Directory, Fondazione Policlinico Universitario Campus Bio-Medico, 00128 Rome, Italy; a.debenedictis@policlinicocampus.it

**Keywords:** nurse–patient communication, nurse–patient relationship, personal protective equipment, COVID-19, qualitative study, experience, interaction

## Abstract

Little is known about which communication strategies nurses carried out and whether the nurse–patient relationship has been altered due to the mandated use of personal protective equipment during the COVID-19 pandemic. This study describes how nurse–patient communication and relationships took place from the point of view of nurses engaged in caring for patients with COVID-19. A qualitative descriptive study design following COREQ guidelines was conducted. Semi-structured telephone interviews with nurses working in the COVID ward of an Italian university hospital were performed between September 2020 and June 2021. Ten nurses were recruited using convenience sampling. One overarching theme, three main themes, and nine sub-themes were identified. The overarching theme ‘The in-out relationship: ‘in here and out there’ and ‘inside me and out of me’ included the main themes ‘A closed system different from normal’, ‘Uncovering meaningful human gestures’, and ‘A deep experience to live’’. The relational nature of nursing—where ‘me and you’ and the context are the main elements—leads nurses to find new ways of interacting and communicating with patients, even in a new situation that has never been experienced. Enhancing human gestures, thinking about new contexts of care, and educating new generations to maintain human-to-human interaction, regardless of the context of care, are the directives to be explored for creating the future of nursing care.

## 1. Introduction

The nurse–patient relationship is at the core of nursing practice, and communication is the means that initiates, elaborates, and ends this relationship [1,2,3,4]. Communication is a basic instrument for providing humanized nursing care to a patient, and effective nurse–patient communication is essential in improving the quality of healthcare [5,6].

Nurses and nursing scientists since Florence Nightingale in the 19th century pointed out the key role of nurse–patient communication for nursing. In the nursing context, communication is a dynamic, complex, and context-related ongoing multivariate process in which the experiences of the participants are shared, and it is based on the exchange of information with the aim of understanding [7].

In the interactive context of the nurse–patient relation, communication is almost never clearly defined or delineated from interaction, and in most cases, the terms are reported interchangeably in the literature [7]. However, some authors suggested how interaction lies at the basis of communication, with interaction as the superior concept and term [8,9].

Peplau describes the nurse–patient relationship as a particular kind of interaction whose components are two individuals (nurse and patient), professional expertise, and patient need [10]. The pioneer in psychiatric nursing claimed how “behavior of the nurse-as-a-person interacting with the patient-as-a-person has significant impact on the patient’s well-being and the quality and outcome of nursing care” [10]. More recent studies have shown how the quality of the relationship directly affects the quality of care provided [11,12].

Within the nurse–patient interaction, the nurse uses verbal and non-verbal communication to establish a meaningful relationship with the patient [1]. Verbal communication, body language, gaze and eye contact, human touch, and facial expression are powerful tools to build a therapeutic, trusting, and comfortable relationship with the patient [13,14,15].

During the COVID-19 pandemic, the care context of patients’ isolation and the use of personal protective equipment (PPE) interfered in interaction and communication with the patient [16,17,18,19]. The mandatory use of PPE, which partially or entirely hid the face and body, placed a physical barrier between the healthcare professional and the patient. PPE prevents the patient from seeing the faces of healthcare professionals and can create a barrier to effective communication [20,21]. The use of a facemask makes it more difficult to recognize non-verbal expressions [20]. Lack of facial recognition becomes a considerable barrier for older adults with cognitive, communication, and/or hearing challenges [22].

Nurses experienced difficulties in providing proper care for patients due to the restrictions created through infection control procedures and uncertainty about infection control [23]. They were not able to enter the patient’s room confidently and were unable to touch patients effectively [24,25]. A cross-sectional study with 120 nurses on the nurse–patient measured interaction during caring for patients with coronavirus disease using the Caring Nurse–Patient Interaction Scale. The results indicated a moderate level of nurse–patient interaction compared to the range of the scale used [26]. The authors attributed this to fear of contamination, workplace environment and workload, available safety measures, and perceived level of knowledge about the pandemic. Simultaneously, a study found that the gaze, the touch of a hand with two pairs of gloves, and words of help or comfort were recognized as the most valuable elements when caring for patients with COVID-19 [27].

Some early studies have suggested interventions to improve patient communication in the current pandemic-related care setting [18,20,22,28,29,30]. However, most of the literature published to date remains at the level of author recommendations, without evaluating whether these barriers affected the nurse–patient relationship and whether nurses adopted alternative communication strategies.

There are currently no studies in the literature that analyze whether wearing PPE has directly and significantly influenced nurse–patient communication and the relationship between them in the recent pandemic experience. Moreover, it is unclear which interventions nurses have adopted to overcome communication barriers and establish an effective relationship with the patient in a context of isolation with a high risk of infection.

Through drawing on the first-hand experiences in the field, which involved nurses with no prior training in caring for a high volume of patients with infectious diseases, this article aims to describe the experiences of nurses in interacting with patients despite communication barriers.

Therefore, the purpose of this study is to explore nurse–patient communication experienced by nurses engaged in caring for patients with COVID-19 and required to wear personal protective equipment (PPE). Specifically, the study aims to achieve two main objectives: firstly, to evaluate from the nurse’s perspective if and how wearing PPE affects nurse–patient interaction; and secondly, to identify the strategies nurses have adopted to foster communication and develop nurse–patient relationships within a COVID ward.

## 2. Materials and Methods

### 2.1. Design

This study was designed as a qualitative investigation utilizing a phenomenological-hermeneutic approach, which involved conducting semi-structured telephone interviews [31,32]. The choice of this approach was motivated by the aim to explore the direct experiences of the study participants regarding the phenomenon under investigation in its natural context [33].

The Consolidated Criteria for Reporting Qualitative Research—COREQ checklist was used as a guideline to report the study data [34].

### 2.2. Setting and Participants

The study was conducted at a 380-bed university hospital in Rome (Italy) in October 2020 and July 2021, involving different nurses during the two phases of the health emergency response.

The university hospital is a private not-for-profit institution that provides healthcare services under the National Health Service. In response to the health emergency, a COVID center was established from April to June 2020 through converting hospital spaces intended for the imminent opening of the Emergency Department, which was postponed to September 2021. The center had 9 intensive care beds and 31 ordinary beds. A second COVID center was opened from October 2020 to June 2021 in response to the second wave to expand the regional health service bed capacity for COVID patients. The spaces of the Emergency Department were again converted into a COVID ward with 14 intensive care beds, 20 semi-intensive beds, and 10 ordinary beds. In addition, a medical department was converted into a COVID ward with 12 semi-intensive beds and 38 ordinary beds. Healthcare professionals from different specialist units were assigned to the COVID department in frontline positions. In both openings, all nursing staff voluntarily chose to work in the COVID wards.

A purposive sampling method was used to select front-line nurses, as nurses providing service directly to patients, who had cared for COVID-19 patients during the first Italian wave in March 2020 or during the second Italian wave in October 2020. The inclusion criteria were the following: (a) be a front-line nurse, (b) have direct experience of caring for COVID-19 patients for at least 4 continuous weeks, and (c) be willing to participate in the study. Nurses who had difficulties with Italian, were newly hired, or were not willing to participate were excluded from the study.

The potential participants were contacted via email by the researcher (RG), who provided information about the study and its objectives, methods, and purpose. The nurses were asked to voluntarily participate in the study and were given the option to withdraw at any time without giving a reason. The nurses were also assured that their information would remain anonymous. The researcher requested a preferred date and time for the interview. An affirmative response to the email was considered as explicit consent to participate in the study.

The sample size was determined based on the principle of data saturation, which was achieved through the snowball sampling method to invite participants [35]. To ensure data saturation, the authors discussed the extent to which new data added value to the existing data [36]. The dominant themes recurred as early as the seventh interview, and no others emerged after it. Accordingly, a total of 10 front-line nurses were enrolled in the study, including 6 who had provided care for COVID-19 patients during the second Italian wave in October 2020 and 4 who had cared for COVID-19 patients during the first Italian wave in March 2020.

### 2.3. Data Collection

Semi-structured, personal, in-depth interviews were conducted among the selected nurses. The outline of the interview was formulated and discussed by four researchers (RG, CDG, ADB, DI) to investigate how the use of personal protective equipment and coveralls influenced the communication in the connection between nurse and patient. The researchers used Peplau’s four components of building the nurse–patient relationship—two individuals (nurse and patient), professional expertise, and patient need—to guide them in formulating the questions [10].

The following open questions were consistently asked:-Could you describe your experience with establishing both verbal and non-verbal communication with patients while working in the COVID ward?-How personal protective equipment affected your ability to communicate with patients? How personal protective equipment affected your relationship with patients?-Was the patient able to interact with you in a complete and profound way? If not, why?-Were you able to share with the patient, even informally, any goals of improving their well-being? Do you remember any examples?

To comply with social distancing measures during the COVID-19 pandemic, interviews were conducted by a trained nursing student (CDG) over the phone, in the participants’ native language (Italian), while they were at home. No one else was present during the interviews, and participants provided their consent at the time of selection and again at the beginning of the interview. The interviews lasted an average of 30 min. The interviews were recorded (CDG) and transcribed verbatim on a Word file within 24 h of each interview. At the beginning of the interview, participants were asked to provide their age, years of work experience, department of origin, and the start and end dates of their work in COVID wards. To ensure anonymity, the trained nursing student identified interviewees using codes such as N1, N2, and so on.

### 2.4. Ethics

The authors’ interest in the research topic was driven by their aim to enhance the care of COVID-19 patients in their institution and provide support to the staff involved in caring for these patients.

As the first author (RG) was a member of the Directory of Health Professions, a nursing student (CDG) was trained to conduct the interviews to ensure that the nurses’ responses were not influenced in any way. To minimize the possibility of coercion, data anonymity was guaranteed, and participation in the study was voluntary. To protect the confidentiality and anonymity of the participants, the trained nursing student removed any identifying information before giving the transcribed interviews to the researchers who performed the analysis.

The University’s Ethics Committee approved the study (Prot: 106.20 OSS ComEt CBM).

### 2.5. Data Analysis

The transcribed texts were anonymized by the interviewer (CDG) before being handed over to two researchers (RG, DI) for interpretation and categorization of the data. The content analysis was performed using the approach proposed by Forman and Damschroder (2008) [37], with each researcher first analyzing the data independently and then reaching a consensus. The researchers thoroughly read the transcripts to gain an overall understanding of each participant’s data before organizing them into discrete units for analysis. The units of analysis were manually coded into the left column of a three-column table in an Excel file, with the central column reserved for codes and the right column for themes. After analyzing the code patterns, the first draft of sub-themes was generated. Investigator triangulation (RG, DI, ADB) was conducted to agree on the themes and sub-themes. Finally, quotations were extracted for each theme to support the credibility of the findings. The emerging themes, sub-themes, and quotations were translated from Italian to English by a bilingual researcher (DI).

## 3. Results

Ten nurses were interviewed, two male (20%) and eight female (80%), with a mean age of 27 years (SD = 5) and an average of four years of experience (SD = 6). No nurses refused to participate in the study, and no interviews were repeated. Three participants self-declared themselves positive for COVID at the time of the interview. None of them had significant clinical conditions that prevented participation in the study. Table 1 shows the characteristics of the nursing staff who were interviewed.

An overarching theme emerged from the content analysis of the interviews: “The relationship between the inside and outside: ‘inside me and out of me’ and ‘in here and out there’”. This overarching theme was supported by three themes and six sub-themes. Table 2 summarizes the themes that emerged from the textual analysis.

### 3.1. The In-Out Relationship: ‘In Here and Out There’ and ‘Inside Me and Out of Me’

A key finding that emerged from the nurses’ interviews was an overarching theme of their experience. Through sharing their experiences, the nurses repeatedly referred to the concepts of “in here” and “out there” in the unique care setting, and “I” and “you” in their interactions with patients.

Nurses reported a clear distinction between two different contexts. They constantly referred to their experience inside the COVID center as a particular physical place, completely different from the outside. The experience in this closed context is perceived as abnormal (in here) compared to a normal one (out there).

N7 raw 49: “*Many (referring to patients) were worried about the fact that outside there the life continued and they had to remain alone in that room, 24 h, everyday, and some even for months, inside there, gazing (staring) at the white wall in front. Putting myself in their shoes....it was really hard for them (patients)*”.

At the same time, what nurses experienced inside them is something distinct and profound of their experience, in addition to what they saw happening outside them. In particular, nurses reported how what they live within themselves (inside me) is distinct from what the patient lives (out of me).

N4 raw 88: “*before, to tell the truth, the other time (referring to the first pandemic wave), I was too focused on myself and how to survive in the difficult physical conditions, the coverall, the foggy glasses, …now (referring to the second pandemic wave) probably, because I knew what to expect, let’s say, I lived it better, I was more out of me, I was more focused on the other*”.

### 3.2. A Closed System Different from Normal

Both patients and nurses live in a distinct context, separate from the ordinary one. Nurses perceive that they live in a different dimension with several new problems to solve. However, while the environment in which patients and nurses exist may seem isolated from the world, a rich network of human connections takes place within it. The shared difficulties seem to increase the bond between those who are living the same experience.

#### 3.2.1. Multiplied Internal Interactions

Within the confined space, human interactions are amplified and diversified, encompassing interactions between patients and multiple healthcare providers, patient-to-patient connections, and interprofessional communication.

N.2 raw 82 “*Wearing coveralls with our names written and all looking the same, nurses, nursing assistants, cleaning staff, has indeed created greater trust from the patients towards the staff, regardless of their role. This enabled us to create a nearly familial environment where the patient knew exactly who to ask for what, while still respecting everyone’s roles. This created a completely different relationship compared to that of a ward*”.

N.5 raw 84: “*There was a patient who exercised every day, so we encouraged him and had all the other patients join in and cheer him on*”.

#### 3.2.2. Facing Multiple Obstacles

The lack of contact with the outside world, the absence of family affection, and the inability to smile at the patient due to the need to wear PPE are obstacles that lead nurses to repeat often “*the only thing you have*” when referring to how to create a caring relationship with the patient.

N.1 raw 25: “*the only way (referring to relationship) is to talk to the patient because the only means of, let’s say, of conveying something to them, is that, is through speaking...They (patients) have only their sense of hearing, no sight or touch, due to the lack of physical contact, all of which is mediated by PPE*”.

#### 3.2.3. Feeling Infectious

The nurse–patient relationship is affected by the contagious nature of the disease, which is a constant concern for nurses in terms of their risk of infection, as well as their consideration for the patient’s feelings about being infectious. The fear of contracting the disease and empathizing with the patient’s experience of isolation shapes the interaction between the nurse and the patient.

N.3 raw 66: “*you cannot have the contact with the patient as you will normally, whether you want it or not, because even if you wear PPE, you have to be cautious about everything you do (…). For instance, I tell you something simple, when we bathe them (patients) in bed, you know, we have to put them in a lateral position and they may be afraid of falling, right? And…Normally, in the ward, they grab onto your uniform. They (COVID-19 patients) tend to do the same thing, but we have to tell them “no don’t touch us, hold onto the bed side rail” because you are anxious about getting scratched by their nails and having the coveralls ripped open, do you understand, and you get contaminated. So, even though it may seem like we are trying to put an end to our relationship with the patient, that’s not the case, it is because, we do it for our own safety*”.

N.1 raw 39: “*The discomfort, …because, being dressed like that, we reminded (the patients), at all times, that they were infected and that we needed to keep the distance…and then, the problem, as they later told us, was that the coveralls and all the other equipment reminded them of, in every moment, the detachment we had to have with them and that they also had with the rest of society*”.

### 3.3. Uncovering Meaningful Human Gestures

Nurses and patients share the same context, which significantly determines the mode of interaction and communication between them. The obscured face of the nurses, the touch mediated by the protective coveralls, the tone of the voice muffled by the face masks, and the inability for patients to read lips have led nurses and patients to find alternative strategies to interact.

#### 3.3.1. Searching for Ways of Interaction

Despite the communication barrier created by protective devices, nurses value those human gestures that make the patient feel their closeness.

N.7 raw 35: “*The caress, for example, was a way of communicating my closeness to the patient, especially since they couldn’t have anyone near them…parents are not allowed to enter the COVID ward, so I was trying to do something that could calm them in a moment of fear. Or leave a dedication on the medication bandage when I renewed it, so that even a simple heart could cheer them up*”.

#### 3.3.2. Enhance the Gaze and the Hearing

The eyes and the gaze become essential for communicating with the patient. Due to the impossibility of lip reading, hearing is amplified as a precious sense to make oneself understood.

N.8 raw 23: *“Communication with the patient at the beginning was very difficult, the coverall, the three pairs of gloves, the face mask, the face shield created an invisible wall between us and the patient, something was essential, however, even before the words, the gaze, the only part of the face that patients could see, and it was almost incredible to discover what a look can convey, many times, they told me, “Today was an exhausting shift. You look really tired; you can see it in your eyes*”.

N.9 raw 19: “*(…) we must yell out to be heard, spell the words well and try to communicate what we wanted to say*”.

#### 3.3.3. The Importance of Holistic Care

Nurses naturally seek to promote and protect patients’ well-being and maintain effective communication with them.

N.3 raw 55: “*practically, this patient, she experienced a panic attack, more than panic attack, it was given by the loneliness, she despite being used to live alone, her children were present in her life, and the fact that, due to the restrictions, parents were not allowed to visit the ward, they (patients) feel the absence strongly. So, one day for example, there was this patient, she asked me “hold my hand, keep me company because it’s good for me to talk”. She was tachypnoic, dyspnoic, but, in that moment while I held her hand, there, I listened to her talk, and she calmed down*”.

Moreover, the human response to the clinical conditions related to COVID seems to clearly emerge from the stories of the nurses: the ability to breathe autonomously or not, the ability to walk or eat, and the need to refer to one’s faith are needs reported alongside the clinical data that characterize the conditions of patients with COVID.

N.1 raw 97: “*...for example, he (patient) was not eating, we tried so many ways to encourage him to eat, so much, that he started eating. We didn’t do anything in particular, just joking and making him laugh. (…). And then, consequently, if you want to write it down, besides improved nutrition, among other aspects, he also benefited on a respiratory level because we managed to convince him to start getting out of bed, the bedridden state ended, and so even in the level of the movement and mobility the solicitation was efficacious*”.

N.4 raw 71: “*The priest had just passed by this atheist patient, he (the patient) holds an image of the Blessed Virgin Mary in his hand, he took that image in his hands everyday, really, and I asked him “do you believe in her?” he said “no I am not a believer, but I like this image”. So, from that moment on, when I saw the priest, I told him to go and have a chat with the patient*”.

### 3.4. A Deep Experience to Live

All of the nurses describe their experience within the COVID department as something unique and incomparable to what they have experienced in other contexts. The uniqueness is not only related to a particular experience from a professional point of view, but to a difficult life experience that is both meaningful and enriching.

#### 3.4.1. The Unique Nurse–Patient Relationship

Constant and frequent interaction creates a special relationship between the nurse and patient. The nurse sees themselves as the privileged person with whom the patient relates. The patient–nurse relationship becomes closer and more meaningful, and the nurse considers themselves as the primary reference for the patient.

N.4 raw 60: “*There were times when I could stay and listen, when I was not overburdened with work, and could sit down, in those moments, I don’t believe there was a limit, and frequently, in fact, I have seen them (patients) cry and communicate their fears, saying “I am afraid”, and yes, who did that (stay and listen to patient), did that till the end, due to the severity of the situation, as if they (patients) needed it, that is I ‘confide in you because I couldn’t see anybody else*’”.

Especially in cases where the family is absent due to external restrictions, healthcare professionals strive to recreate a sense of family around the patient.

N.2 raw 73: “*so, one thing that was not particularly pleasant, it was providing assistance to patients who were not well and nearing the end of their lives, when they were dying, so, we tried to create an even stronger support system for them and replace, as much as possible, the affection they would have received from their families, in this way, we tried to care for them even better than we would with a ‘normal’ patient who has the support of close family members. We attempted to recreate that sense of familial affection by staying close to the patient and waiting with them through the worst moments*”.

At the same time, in a closed context with very few or no relations with the outside world, nurses’ attention appears to be more focused on patients’ needs, and the nurses themselves become more aware of the value of their care.

N.8 raw 59: “*I should say that the emotions, tears, sweating, tiredness, and laughter that the COVID ward gave me, made me understand the true meaning of nursing care…*”

N.6 raw 55: “*I must say that, in any case, COVID leaves you alone in a hospital bed, and so, I hope that all patients have felt my presence and closeness, even if only through a glance*”.

#### 3.4.2. Transmitting One’s Own Experience

Narrating one’s life and past experiences in the context of their present condition becomes a spontaneous way for patients to strengthen their relationship with the nurse.

N.4 raw 67: “*In this particular moment, they (patients) come to understand, that there is, the situation is serious, really I think that… One has even confided in me about his life, such as feeling guilty about something that happened years ago with his brother*”.

N.6 raw 41: “*One time, in particular, I remember when a couple—a husband and wife—were both hospitalized, in the same room. The husband was dying because of COVID-19, and the wife was telling me regarding the motif…how he got infected, things that could have been avoided, and feeling guilty about why they had gotten sick. She was more worried about her husband than herself, … we spent days discussing this*”.

#### 3.4.3. Over Time

Time is an important variable that nurses report in various aspects. The experiences of nurses evolve from a “*before*” to an “*after*” situation.

N.8 raw 41: “*At first, I felt bad, I felt scared, and unsure of what might happen to me, I didn’t know what to say them (patients). I struggled to find the right words to say to them (patients), because sometimes the words don’t come out, …but as the days went by, I grew stronger. I was able to find words of comfort for each of them*”.

Patients and nurses have different perceptions of time within the COVID ward.

N.10 raw 25: “*After a while, a sense of depression set in among the patients because everything appeared the same to them round the clock, making it difficult to distinguish between day and night*”.

Nurses often recall past events to gather experiences from memory. Phrases such as “*it comes to mind*”, “*by heart*”, and “*remembering*” are frequently used to report their experiences, suggesting that memory is the only guide for action in new situations, especially in the absence of other experiences.

N.2 raw 60: “*I remember an episode that comes to my mind, we had a foreign patient with whom communication was difficult. We tried to convey to him the importance of mobilization (…)*”.

Memory also contains intense emotions that are difficult to forget.

N.9 raw 35: “*Over the course of six months, I have heard many stories. What I regret the most is that many of them (patients) did not survive, and I carry the weight of their stories with me. The story that struck me the most, although all of them left a mark on me, was the memory of a man, a father… This gentleman did not make it and unfortunately he lost his battle with COVID-19. I will remember him forever*”.

## 4. Discussion

The recent pandemic has placed nurses’ actions at the center of the response to the care needs created for most of the world’s population. Nurses involved in direct assistance to COVID-19 patients, in a climate of emergency with limited knowledge of the disease and high volumes of complex cases to handle, experienced a new and profoundly significant form of care delivery with the constant risk of contagion.

This study explores the nurse–patient interaction and communication experienced by nurses engaged in caring for COVID-19 patients while required to wear protective coveralls. The overarching theme of “*the in-out relationship*” expresses the simultaneous concepts of “*in here and out there*” and “*inside me and out of me*”. This highlights how, even in conditions that seem to impede the care relationship, the basis of this relationship is the “*I*” and “*you*” living in the same context [38,39].

The overall results of this study confirm the influence of caring for COVID-19 patients on the main paradigms of nursing care. Patient care in this context is characterized by the experience of death away from home, the role of conduit for family members during illness and hospitalization, and proximity between the patient and nurse. Despite greater communication barriers in new scenarios, the fundamental assumptions of nursing remain intact, as shown in previous studies [27,40,41]. Moreover, in a closed environment with professionals and patients sharing the same space for prolonged periods, interactions between people seem to multiply, and teamwork strengthens. However, as reported by a recent study, intentional efforts by nurses to create an interactive space are needed to provide the necessary care [42].

Among the many problems that healthcare professionals have faced, the infectious context poses a prominent barrier that exacerbates the difficulty of communicating and relating to the patient. The risk of becoming infected has been identified as one of the most significant communication barriers that nurses face when caring for patients with COVID-19 [23,43]. However, contagiousness is not only a personal risk but also a reminder that healthcare workers have to constantly wear PPE while providing assistance. Many experiences in the field have shown that overcoming this barrier has also led healthcare workers to perform heroic acts, putting their own safety at risk due to the lack of adequate protective devices [44,45,46,47]. Reflection on the value of helping a fragile person, a value perceived as superior to others, should continue to shape nursing and the way of carrying out a caring relationship [48,49,50].

Through contributing to a reflection on humanizing care, even in care settings perceived as exceptional, the research results demonstrate that nurses are inclined to seek a relationship that includes essential gestures in care. The use of PPE significantly interfered with face-to-face nurse–patient communication at the bedside. However, when the usual ways are restricted, there is a rich potential for interacting and communicating with the patient. Nurses reported that communication is valuable not only for the content but also for the need to transmit, to be close, and thus to create a meaningful relationship with the patient. For this reason, non-verbal communication emerged more forcefully through gaze and gestures to establish a connection with the patient despite the barriers of the coverall. These results are comparable to what is reported in the study by Sugg et al. (2023) [41]. Touch, hand-holding, writing information, using pictures, gesticulation, and body language represented a new and perhaps more effective way of interacting with the patient, who was aware of the difficulty posed by the context.

Each person knows and relates to their surroundings through their body, using the five senses. Indeed, the body is the human way of knowing reality, and nursing is based on a body-to-body care relationship: a caress, a look, and wiping away tears are actions that humanize care and cannot be excluded from the care of a person [51]. Where PPE makes it difficult to touch the patient’s skin, nurses seek other ways to enhance the gaze and continuously use communication feedback.

Furthermore, this research highlights that in the context of isolation and without being able to relate to their loved ones, the patient’s need for connection emerges with more force. A recent review by Fernandes et al. (2022) [52] cites several interventions to humanize care for patients affected by COVID-19 in isolation units, including adopting communication strategies to establish an interpersonal relationship between nurses and patients, providing physical and psychological comfort, sharing in decision-making, providing patient education, and managing patients’ symptoms. As the interaction between nurses and patients takes longer than with other professionals, nurses are in a pivotal position to satisfy this human need.

Understanding the impact of this pandemic event on the patient relationship could provide a framework of knowledge to re-evaluate how to make the relationship with the patient meaningful in this new context and bring out what makes a relationship human in an era of rapid technological development.

The results of this research show how a subject performing a gesture and the reason they perform it could compensate for the lack of skin-to-skin contact due to the mediation of PPE. Along the same lines, the emotional Italian experience of the “room of hugs”, where patients and family members could hug each other through transparent dividing sheets, highlights the importance of human contact. However, in an era where technology is rapidly advancing with highly technical environments, innovative tools for treating pathologies, and applications of artificial intelligence in healthcare, deeper reflection is needed to emphasize the value of body-to-body relationships in patient care and the added value of nursing [14,53]. The relationship of human touch, without the mediation of technologically advanced medical devices, could become the most critical need of people. We should continue to ask ourselves what gestures humanize care and what makes a caress human in this new era of advanced technology.

Reflection on the multifaceted aspects of communication can uncover unused tools to enhance the nurse–patient relationship in all care contexts. This pandemic posed a challenge to the nursing profession, as it had to develop alternative ways of effective communication to maintain a good care relationship despite the context of isolation. In particular, the sudden and significant involvement of most nursing staff in the health emergency meant that professionals had to draw upon their own experience in the field without any preparation [54]. The results of this research highlight how it is possible to face new situations while maintaining the fundamental assumption of nursing [55]. Nurses build their skills on their own experiences and are strongly motivated to reanalyze all their actions to learn what worked and what did not. Therefore, the memory of what happened is not only a desire to preserve a meaningful experience [27], but it also becomes the only tool to face a new and unexpected situation with a complete lack of experience. However, as reported by De Benedictis et al. (2022) [39], the uniqueness of the lived experience and the motivation and sense of mission that emerged from nurses may have contributed to activating superior forces to respond to a crisis that involved ethical, professional, and personal aspects of caring.

This study provides valuable insights that can inform strategies and actions for equipping nurses with the necessary tools and resources to provide care for patients in infectious care settings. Through simulating ordinary care activities in a different care context, the creativity of those who work in the field emerges to find effective solutions to prevent and control infections. The simulation should be focused on how to maintain communication and relationship with the patient, as well as on the different and more complex procedures to be performed. Moreover, it is important to anticipate and prepare for future public health emergencies through providing training to staff on maintaining the patient care relationship in this unique and challenging context. In addition, when faced with emergency situations where knowledge and skills may be lacking, the training of healthcare professionals should focus on strengthening teamwork and facilitating the rapid exchange of experience and knowledge gained from working in the field. Helping one another should be a priority skill to acquire.

Despite the results achieved, this research has three significant limitations in methodology that do not make the results generalizable. First, the sample consists of only 10 nurses belonging to the same hospital. Given the novelty and immediacy of the pandemic event and its strong impact, it was essential to collect data in the field at the time they were experienced or shortly after, also considering the objective limitations in conducting field research during an emergency phase. As a matter of fact, it is important to keep in mind the difficult and complex context in which this research was conducted and the simultaneous management of the pandemic, which both researchers and participants managed firsthand in a shortage of time and physical locations to conduct in-depth research. For this reason, the choice of methodology, i.e., telephone interviews, was strongly influenced by a context in which it was difficult to have a direct relationship with professionals caring for COVID-19 patients. In addition, during the first phase of the pandemic, it was difficult for students and unlicensed staff to enter the hospital. However, focusing on the specific topic of nurse–patient communication enabled participants to bring out significant aspects precisely related to this unique context. The richness of the themes that emerged led the authors to consider the collected data of value and significance, also in the light of the studies published relating to the same period and with the same sampling limitations [56]. In this regard, a very interesting recent study reports how methods and tools of evidence-based medicine were designed primarily to answer simple, focused questions in a stable context [57]. The complex questions derived from the pandemic context brought out its significant limitations, and further studies can be carried out on how to capture data in complex and emergent contexts as they come up.

Second, the short duration of the interviews may have somewhat limited the emergence of additional or contrasting themes. Again, the focus of the questions on nurse–patient communication actually limited the answers, which nevertheless presented a particular richness in terms of content, examples, and meanings.

Finally, due to the purposive recruitment method, the selection of nurses most likely to accept participation in the study may not have considered additional important aspects not mentioned by them. Moreover, the unrepresentative sample of predominantly white Italian participants in a specific context with adequate resources for the care of COVID-19 patients may have highly influenced the responses. The cultural context, nursing training, and the resources available in every care setting make it difficult to compare and generalize these results. However, different experiences gathered in all contexts can form the basis for further research on communication and the care relationship with the patient.

## 5. Conclusions

Nurses are at the forefront of fighting the pandemic for both the considerable amount of time they spend in the face-to-face nursing of COVID-19 patients and for the procedures put in place to counter the spread of the virus. Indeed, at the time of this writing, nurses are still responsible for providing holistic care for a significant volume of patients with infectious disease in a context of isolation and risk of contagion. This very new and particular context of care globally experienced by nurses has made it possible to highlight essential aspects of the essence of nursing.

In this study, nurses’ ability to interact and communicate with patients despite barriers was revealed as a core element in developing the nurse–patient relationship as a caring relationship. Communicating with all the senses, particularly with the gaze and gestures, maintains the human relationship that is the basis of care. Moreover, in a context of isolation, human gestures of care seem to be a fundamental factor for both those who care and those who are treated. However, in a world increasingly focused on high-tech treatments, further research is needed to gain a greater understanding of what makes a gesture human and if human-to-human contact is an essential element in patient care.

## Figures and Tables

**Table 1 healthcare-11-01960-t001:** Nurses characteristics (N = 10).

Gender	N	
Male	2	
Female	8	
**Department of origin**	**N**	
Medicine	2	
Surgery	2	
Oncology	1	
COVID (first work experience)	5	
	**Average**	**SD**
**Age**	27	5
**Years of experience**	4	6
**Days of experience in the COVID ward**	110	64

**Table 2 healthcare-11-01960-t002:** Overarching theme, themes, and sub-themes emerged from the content analysis.

**The in-out relationship:** **‘in here and out there’ and ‘inside me and out of me’**	**A closed system different from normal**
	Multiplied internal interactions
	Facing multiple obstacles
	Feeling infectious
	**Uncovering meaningful human gestures**
	Searching for ways of interaction
	Enhance the gaze and the hearing
	The importance of holistic care
	**A deep experience to live**
	The unique nurse–patient relationship
	Transmitting one’s own experience
	Over time

## Data Availability

Data are unavailable due to privacy and ethical restrictions.

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
