# Peer review of "Nurse–Patient Communication and Relationship When Wearing Personal Protective Equipment: Nurses’ Experience in a COVID-19 Ward"

_healthcare, 2023, doi:10.3390/healthcare11131960_

Round 1
Reviewer 1 Report
In my view, this article is well written and provides a detailed and impressive insight into the experiences of nursing staff during the Covid-19 pandemic. I have only a few comments.
Introduction
p.1/2, second and third paragraph: According to the references given, the statements regarding communication refer to the nursing context/the interactive context of the nurse-patient relation. It may be useful to mention this once again as some readers may be unsure whether this is about communication in general.
p. 2, line 49: Please write “describes” instead of “describe”.
p. 2, line 66: Please write either “facemasks” or “a facemask”.
p. 2, line 68: Please consider revising the term “hard barrier”, this sounds a bit odd.
p. 2, line 81: Please add “pandemic-related” or similar when referring to the “current” care setting.
Method
2.2 Setting and participants
p. 3, line 128: Not every reader is familiar with the term “frontline nurse”. Please consider providing a brief explanation.
2.3 Data collection
p. 4, lines 156/157: Please revise the sentence (“How did PPE affect…”).
p. 4, lines 167 et seq.: Can the data collected really be seen as anonymized? Given the socio-demographic data assessed and the use of codes (N1 etc.), is it not rather a case of pseudonymization?
2.5 data analysis
p. 5: Has any kind of statistical testing of the investigator agreement/concordance been performed?
Results
p. 5, table 1, upper part: Given the small sample size, it should be sufficient to report absolute numbers, percentages are not necessary.
Discussion
p. 11, lines 484 et seq.: It would be interesting to learn which specific content the training of nurses (and other healthcare professionals) might address beyond the key points mentioned by the authors.
Reviewer 2 Report
Review
Nurse-Patient Communication with PPE
ABSTRACT
The abstract describes the study well.
INTRODUCTION
A cross-sectional study with 120 nurses reports a moderate 73 level of nurse-patient interaction during the care of patients with COVID-19 [27]
This sentence was not clear – was interaction reduce or the same during the pandemic? What is a moderate level?
The study has 2 clear objectives:
firstly, to evaluate from the nurse's perspective if and how wearing PPE affects he nurse-patient interaction;
and secondly,
to identify the strategies nurses have adopted 98 to foster communication and develop the nurse-patient relationship within a Covid ward.
MATERIAL AND METHODS
The study was a qualitative investigation.
The COREQ checklist was used.
There was a good description of the Covid-19 wards where the research was conducted.
Purposive sampling was used of nurses included in the study with clear inclusion criteria. Snowball sampling methods were also used.
Sample size was determined based on the principle of data saturation.
An outline for semi-structured interviews was developed including utilisation of theory (Peplau’s 4 components).
Demographics of the participants was appropriately recorded.
The study was conducted with appropriate Ethics, and had approval for this from a University Ethics Committee.
RESULTS
Ten nurses were interviewed.
There was an overarching theme of : between the inside and outside: 'inside me and out of me' and 'in here and out there'".
The over-arching theme was supported by three themes and six sub-themes – see Table 2.
It was fascinating to hear nurses were able to change focus from an inward focus to being able to care for others as they became more familiar with how to manage the Pandemic over time.
There was a description of paradoxical amplification and diversification of human interaction.
There was appropriate use of representative quotes.
The results section concludes with a section on the importance of memories and emotion, especially in the context of patients who received end of life care.
DISCUSSION
There was good discussion, using the findings of the study, of how interactions with patients multiplied, and teamwork was strengthened. This was despite the barriers that Covid-19 infection presented between patients and nurses.
The student highlights humanizing care, and the inclination of nurses providing relational care with appropriate gestures.
“Touch, hand-holding, writing information, using pictures, gesticulation, and body lan- 431 guage represented a new and perhaps more effective way of interacting with the patient, 432 who was aware of the difficulty posed by the context”.
There is good integration with other literature e.g. Sugg et al. (2023) and Fernandes et al. (2022).
The discussion finishes with appropriate acknowledgement of limitations. This included the purposive recruitment method – and representativeness of the sample (predominantly white Italian participants).
Are there other methods that could be employed e.g. ethnographic observation or conversation analysis (ie video recording)?
CONCLUSION
This is a good summary of the study.
Consider including the over arching theme and 3 sub-themes into the conclusion.
Reviewer 3 Report
This study brings a very important topic to the fore, that is, how communication from and by nurses, during the pandemic, may have been affected by the pandemic's protective equipment. It proposes a qualitative study design to do so. Ten nurses were recruited to participate in telephone-based interviews.
I think this is a promising topic of study and publication. The topic is indeed extremely important and many of the ideas presented in the paper are extremely valuable. there is also a dire need for this type of study that explores the 'human/affect' impact of the pandemic on care practices, that is, nurses-based ones, and on creating a rapport with patients.
I do not think however, that this study is sufficiently scientifically robust for publication. There is some concerns that I have as well, which I explain below. I would like to see a revised copy of this publication, but further work is required. I give some suggestions as to how this could be achieved.
**
There are, in my view, many concerns, all regarding the study design, methods used, as well as regarding how the results are presented. I provided my notes-while-reviewing below, with further explanations for each point; but here is the list of my concerns:
- inadequate sample size and diversity: too small, all in the same hospitals; length of interviews, too short; inaccurate description of the methodology used given the sample size, methods used and length of interviews (i.e. interpretative-phenomenology).
- uncertainty regarding the quotes -- they all are expressed in the same voice. Effect of the translation? It's hard to believe this level of richness was provided in 12 to 20 minutes interviews. Did the participants receive the interview questions in advance?
**
Introduction/lit review: Overall, I found the introduction/context and discussion sections the strongest, excellent well written, very accurate, informative, intelligent. At at theoretical level, this is very rich.
Methods:
what approaches/method/lit review were used/done to assess that there aren't any other studies on this topic? Doesn't seem right... I'd be surprised to learn it is the case.
Please include the length of the interviews in the methods (not results).
Please indicate why phone-based interviews was chosen (rather than virtual or in person).
Could you provide justification for the sample size. In my view, this seems extremely low.
Also, the data collection includes only one hospital. I do not think this is sufficiently, scientifically. It is exploratory, yes, but in order to be scientifically sound, it would require comparision with existing literature/other studies, at a minimum; or a larger sample size, with at least 3 sites
-I'm not sure this level of granular details is needed for the publication, i.e. lines 190-191
I found many issues with the scientific validity and value of this study.
In addition to the issues already specified above, the length of the interview is greatly inadequate and insufficient, and I do not think the authors can speak of a 'interpretative-phenomonology approach' to the data collection, with 12 to 20 minutes interviews with 10 individuals.
I also question how three interviews were conducted while the person was sick with COVID. We are not sure how severe or worrisome the symptoms and experience of COVID could be and how this could influence the interview's responses. There was only one interview conducted, which also is insufficient in the context of the methodology proposed.
Results: the analysis of the interviews is also very rich -- but given (1) the sample size and (2) my concerns regarding the accuracy of the quotes selected, I recommend that authors explain a little more why the quotes appear all written in the same voice; how this level of richness was achieved -- with all participants! -- in 12-20 minutes interviews.
Please review the numbering of themes
Discussion: Very interesting discussion. I find, however, that given the context of the study (sample size, methods, etc.), it would require a more comparative component with existing literature, for scientific validity and robustness
Limitations: it is not 'limitations' if you justify them as, in the end, not limitations. Limitations are limitations, period.
Conclusion: Very good
Round 2
Reviewer 3 Report
I am fine with the responses from the authors and the revisions provided. Thank you.
There are some minor typos; please do a thorough read-through